# Cold Gas-Dynamic Spray for Catalyzation of Plastically Deformed Mg-Strips with Ni Powder

**DOI:** 10.3390/nano11051169

**Published:** 2021-04-29

**Authors:** M. Sherif El-Eskandarany, Naser Ali, Mohammad Banyan, Fahad Al-Ajmi

**Affiliations:** Nanotechnology and Applications Program, Energy and Building Research Center, Kuwait Institute for Scientific Research, Safat 13109, Kuwait; msherif@kisr.edu.kw (M.S.E.-E.); mbanyan@kisr.edu.kw (M.B.); ftajmi@kisr.edu.kw (F.A.-A.)

**Keywords:** catalyzation, cold-spray technology, cycle-life-time, de/rehydrogenation kinetics, severe plastic deformation

## Abstract

Magnesium hydride (MgH_2_) has received significant attention due to its potential applications as solid-state hydrogen storage media for useful fuel cell applications. Even though MgH_2_ possesses several attractive hydrogen storage properties, it cannot be utilized in fuel cell applications due to its high thermal stability and poor hydrogen uptake/release kinetics. High-energy ball milling, and mechanically-induced cold-rolling processes are the most common techniques to introduce severe plastic deformation and lattice imperfection in the Mg/MgH_2_. Furthermore, using one or more catalytic agents is considered a practical solution to improve both the de-/rehydrogenation process of MgH_2_.These treatments are usually dedicated to enhance its hydrogen storage properties and deduce its thermal stability. However, catalyzation of Mg/MgH_2_ powders with a desired catalytic agent using ball milling process has shown some disadvantages due to the uncontrolled distribution of the agent particles in the MgH_2_ powder matrix. The present study has been undertaken to employ a cold gas-dynamic spray process for catalyzing the fresh surfaces of mechanically-induced cold-rolled Mg ribbons with Ni powder particles. The starting Mg-rods were firstly heat treated and forged 200 times before cold rolling for 300 passes. The as-treated ribbons were then catalyzed by Ni particles, using cold gas-dynamic spray process. In this catalyzation approach, the Ni particles were carried by a stream of Ar gas via a high-velocity jet at a supersonic velocity. Accordingly, the pelted Ni particles penetrated the Mg-substrate ribbons, and hence created numerous micropores into the Mg, allowed the Ni particles to form a homogeneous network of catalytic active sites in Mg substrate. As the number of coating time increased to three times, the Ni concentration increased (5.28 wt.%), and this led to significant enhancement of the Mg-hydrogen storage capacity, as well as improving the de-/rehydrogenation kinetics. This is evidenced by the high value of hydrogen storage capacity (6.1 wt.% hydrogen) and the fast gas uptake kinetics (5.1 min) under moderate pressure (10 bar) and temperature (200 °C). The fabricated nanocomposite MgH_2_/5.28 wt.% Ni strips have shown good dehydrogenation behavior, indicated by their capability to desorb 6.1 wt.% of hydrogen gas within 11 min at 200 °C under 200 mbar of hydrogen pressure. Moreover, this system possessed long cycle-life-time, which extended to 350 h with a minimal degradation in the storage and kinetics behavior.

## 1. Introduction

### 1.1. Hydrogen Energy

Hydrogen as an energy carrier has a lot of potential as a new green energy alternative in the future [1,2,3]. Hydrogen storage, which plays a critical role in establishing a hydrogen economy, has been the focus of extensive research for several years [4,5]. For stationary and portable applications, high-density hydrogen storage is a concern, and transportation applications remain a major challenge [6,7]. Hydrogen may be stored either in gaseous state under 350–700 bar or as liquid phase at −253 °C. In terms of security, metal hydrides are extremely pyrophoric and the cost of the material would be an issue for large-scale applications. In comparison with the traditional hydrogen storage technologies, metal hydrides, in particularly MgH_2_ hold very attractive properties. Where the hydrogenation process can be successfully achieved at 1 MPa, one drawback of Mg is the application of high temperature (Table 1). MgH_2_ system shows a higher gravimetric energy when compared with compressed-H_2_, but a closed value to liquid-H_2_ (Table 1). Moreover, MgH_2_ possesses a higher volumetric energy when compared with the compressed-, and liquid-H_2_. the Table 1 presents a comparison between the three approaches that are used for hydrogen storage.

### 1.2. Magnesium-Based Hydrogen Storage Materials

In contrast to the traditional ways of hydrogen storing, magnesium (Mg) and Mg-based materials have been regarded as the most promising hydrogen storage materials for real-world applications due to their appealing chemical and physical properties [9,10,11,12,13,14,15]. As shown in Table 1, elemental hcp-Mg has a remarkable combination of high gravimetric and volumetric hydrogen storage capacities, as well as excellent cyclability [16,17]. Unfortunately, tetragonal-MgH_2_ (b-phase) is thermodynamically very stable compound (H_for_ = −75 kJ/mol·H_2_) that decomposes at a high temperature of 350 °C [18], and possesses a high apparent activation energy, E_a_ (above 130 kJ/mol) [19]. In addition, this hydride phase has a very slow hydrogen uptake/release kinetics below 350–400 °C [20]. To be used in fuel cell applications, MgH_2_ must go through a series of intensive and serious treatment processes to improve its fundamentally weak kinetic activity and lower its activation energy of decomposition [21].

#### Scenarios Used to Enhance the Hydrogenation Performance of Magnesium Hydride

Mechanical treatments

Several applicable scenarios for enhancing the hydrogen storage behavior of Mg/MgH_2_ have been suggested over the last three decades. Extreme plastic deformation (SPD) [22], which was proposed using a high-energy ball milling (HEBM) technique [23], has shown a substantial improvement in Mg/MgH_2_ powders over a long milling period (>100 h) [23].

β-tetragonal MgH_2_ (most stable phase) powders were gradually disordered and transformed into a less stable phase (γ-orthorhombic) upon increasing the high-energy ball milling time [24,25]. Cold-rolling (CR) [26,27], equal channel angular pressing (ECAP) [28], and high-pressure torsion (HPT) techniques [29] of bulk Mg metal also showed significant results for the formation of nanocrystalline Mg/MgH_2_. In these processes, the introduction of high-intensity defects such as plastic deformation, lattice and point defects, and dislocations, lead to an increase in the volume fraction of nanocrystalline grains. It is realized that the presence of these defects creates nucleation points for hydrogenation, where a large number of grain boundaries assists in fast diffusion pathways for hydrogen [30]. Wagemans et al. [23] have pointed out that the grain size refining of Mg/MgH_2_ powder particles to the nanolevel is a powerful strategy to deduce the high thermodynamic stability of MgH_2_.

Catalyzation

Another common method for improving the hydrogenation/dehydrogenation kinetics of MgH_2_ is to dope with catalytic agents [31]. Since the 1990s, wide range of pure transition metals (TM) in different concentrations have been used to enhance the hydrogen storage behavior of Mg/MgH_2_. Many of these metallic catalytic agents showed remarkable beneficial effects on changing the hydrogen storage properties of Mg/MgH_2_ due to their superior hydrogen splitting (dissociation) and recombination capabilities. Metallic catalytic agents of Ni, Ti, V, and Nb metals [2,32,33,34], as well as their alloys, such as TiV [35], CrTi [36], TiMn_2_ [37], VTiCr [38], and ZrNi_5_ [39] are some of metallic catalytic agents, successfully used to minimize the formation and decomposition temperatures of Mg/MgH_2_. Additionally, as pointed out in our recent studies, doping MgH_2_ powder with metastable alloys, e.g., big-cube Zr_2_Ni [40] metallic glassy alloys, such as Zr_70_Ni_20_Pd_10_ [3], result in an outstanding improvement in the hydrogen storage properties of MgH_2_ powders.

In addition to the metallic catalytic agents, compounds and metastable modifier agents, metal-oxides (e.g., Nb_2_O_5_ [41], Cr_2_O_3_ [41], TiO_2_ [42], and La_2_O_3_ [43]), -carbides (e.g., SiC [44], and TiC [45]), -hydrides (e.g., TiH_2_ [11], LaH_3_ [46], and NbH [46]), and carbon-based nanomaterials such as carbon nanotubes [47], graphene, and nanofibers [48] have been successfully used to enhance the hydrogen storage behavior of MgH_2_.

Drawbacks of high-energy ball milling

The most efficient and widely used technique for doping Mg and/or MgH_2_ powders with the desired catalytic agents in the form of powders/nanopowders/nanoparticles/nanotubes is the mechanically induced doping using high-energy reactive ball milling (RBM) under pressurized hydrogen. During the early stages of RBM (6 h), Ni powders (Figure 1a) appeared to agglomerate to form dense layers embedded in the Mg/MgH_2_ matrix, as shown in the current analysis (b).

Formation of large of Mg/Ni aggregated occurred due to local heat produced by ball-powder-ball collisions during the milling process, which resulted in cold welding/rewelding of the powders. Shear stresses were produced as the RBM time (12.5 h) was increased, resulting in the refinement of the agglomerated thick Ni layers to thin-intimated layers, as shown in Figure 1c. In this process, the Ni layers were disintegrated into coarse powders (Figure 1d) after 50 h of RBM and then polished into smaller particles (0.25–0.78 m in diameter) after 100 h of milling, as shown in Figure 1e. After a long-term of milling (100 h) Ni-particles still showed heterogeneous distribution in the Mg/MgH_2_ matrix (Figure 1e). Accordingly, the MgH_2_/Ni nanocomposite powders of the end-product may widely vary from particle to particle and within the individual powders, as indicated in Figure 1e. 

After 50 h of RBM, the Ni layers were disintegrated into coarse powders (Figure 1d) and then refined into smaller particles (~0.25 to 0.78 μm in diameter) after milling for 100 h, as illustrated in Figure 1e. Some limitations of the mechanically-induced doping with RBM process are the long processing time (100 h) and the heterogeneous distribution of Ni-particles in the Mg/MgH_2_ matrix (Figure 1e). The formation of such heterogeneous MgH_2_/Ni powders leads to the production of large MgH_2_ zones lacking the catalytic agent (Figure 1e).

### 1.3. The Aim of the Present Work

The aim of this study was to investigate the possibility of utilizing a cold spraying (CS) technique to catalyze the Mg-substrates ribbons with Ni powders. It should be emphases that cold spray method, which entered the thermal spraying coating process in the 1980s, is primarily used for surface coating applications to protect metal equipment from oil and water-mediated corrosion/erosion [49,50]. In addition, it has been recently employed for fabrication of useful antibacterial and antimicrobial coatings dedicated for medical and food applications [51]. Furthermore, the current work explores the process of Ni-cold spray coating on the hydrogenation behavior of Mg-surface, as well as the efficiency and performance of cold-spraying system for preparations of Mg/Ni nanocomposites and its impact on enhancing the hydrogen storage behavior. Moreover, the effect of mechanical treatments on the hydrogen storage characteristics of plastically deformed Mg before and after cold spraying with Ni powders, such as forging, cold rolling, and warm pressing, were investigated in this study. It will also be explored how to dope Mg substrates with supersonicated Ni powders. 

Finally, the results will be presented and discussed in terms of crystal structure, morphology, kinetics, and thermal stability. Further studies with different cold-sprayed catalytic agents, including different metals, metal alloys, metal compounds (e.g., oxides, and carbides), composites and metastable alloys (e.g., amorphous and metallic glasses), which are required to establish this proposed catalyzation process will be undertaken shortly.

## 2. Materials and Methods

### 2.1. Sample Preparations

Commercial Mg-rods (90 cm long × 0.8 cm diameter) with a purity of 99.9 wt.% were snipped into 6 short rods of 15 cm in length and then manually forged at 400 °C, as displayed in Figure 2a,b. The as-forged strips were then mechanically deformed, using common cold rolling for 300 passes (Figure 2c,d). The samples were then warm pressed at 200 °C for 5 min after each 10 passes using a two-plate warm press (Figure 2e) to overcome the undesired brittleness that is usually resulted upon cold-rolling step.

After 200-forging cycles at 400 °C, the as-received Mg-rods (Figure 2fi) were elongated by approximately 112%, as shown in Figure 2fii. The average thickness of the -Mg strips after 300 times of cold rolling (CR) was 115 mm (Figure 2fiii). The CR Mg-strips were then cut into short ribbons with a thickness of 0.5 cm and a length of approximately 8 cm (Figure 2g). The strips were simply washed with acetone and ethanol before being dried overnight in a vacuum oven at 150 °C. A certain number (7 pieces) of as-cold rolled Mg-ribbons were firmed onto a tool steel plate and aligned in parallel using paper clips (Figure 3a). The system was then fixed vertically at a cold spray sample stage using two jaw pullers (Figure 3b). Commercial fine Ni powders (99.9 wt.% purity, 1 μm in diameter) selected as the feedstock coating materials were charged in open air into a powder feeder (Figure 4a,b).

Pressurized argon (Ar) was used as carrier gas to deliver the Ni powders from the feeders to the gas jet (Figure 3b and Figure 4a,b) through a reinforced polymer tube, which is illustrated in Figure 4a. This flow was accelerated in velocity upon passing through a heating system, which is illustrated in Figure 2b. The applied pressure and temperature were 6 bar and 150 °C respectively, and the gas flow rate was 275 L/min. The stream (Ar gas + Ni particles) was injected with subsonic velocity into the inlet section of DeLaval-type nozzle (Figure 4b). Once the subsonic flow left the inlet zone toward exhaust expansion zone through a nozzle throat (Figure 4b), its velocity was dramatically increased to 500 m/s. During the cold spray process, the Ni particles were supersonically pelted into the Mg-strips by the compressed Ar gas (Figure 3b and Figure 4b), and the Ni particles flying with this supersonic velocity tended to penetrate the Mg strips.

The CS process was applied to coat each face of the ribbons one, two and three times to study the effect of the number of coatings with Ni particles. After 3 rounds of cold spraying with Ni powder particles, the average thickness obtained upon measuring of 7 samples was significantly increased and was 171 μm. The as-coated ribbons (Figure 3c) were wrapped in trace paper and inserted between two-jaw-type vise for 16 h (Figure 3d). The coated samples were then CR 10 times to ensure a straight structure.

### 2.2. Sample Characterizations

#### 2.2.1. Structural Analysis

X-ray diffraction (XRD) technique, using SmartLab-Rigaku, Tokyo, Japan with CuKα radiation (0.15418 nm) was used to investigate the general crystal structure of the as-cold rolled and cold sprayed strips. The local structure of ultrathin microtome-sectioned bulk samples were examined by 200 kV-JEOL-2100F (Kawasaki, Japan), field emission high-resolution transmission electron microscope (FE-HRTEM).

#### 2.2.2. Morphology

The morphological characteristics of cold rolled and cold sprayed samples were investigated using a 15 kV-JSM-7800F (Kawasaki, Japan) field-emission scanning electron microscope (FE-SEM) JEOL-Japan. The local elemental analysis was performed using an energy-dispersive X-ray spectroscopy (EDS, Oxford Instruments, Abingdon-on-Thames, UK) system interfaced with the FE-SEM.

#### 2.2.3. Thermal Analysis

The hydrogenation and dehydrogenation behaviors of the samples at high hydrogen pressure (30 bar) were investigated using a SENsys-EVO high-pressure differential scanning calorimeter (HP-DSC) from (HP-DSC) Setaram Instrumentation-France. Meanwhile, the decomposition behavior and thermal stability of the samples were examined under 1 bar of He using DSC (Shimadzu Thermal Analysis/TA-60WS, Sitama, Japan). 

#### 2.2.4. Hydrogenation/Dehydrogenation Kinetics

Sievert’s method using PCT Pro-2000, Setaram Instrumentation, Nice, France, was employed to investigate the de/rehydrogenation kinetics of the samples. These experiments were performed at a hydrogen gas pressure of 8 bar and 200 mbar for hydrogenation and dehydrogenation at 150 °C and 200 °C, respectively. The XRD technique was used to examine the crystal structure of the sample obtained after hydrogen absorption and desorption experiments. Six individual groups of Mg-based materials prepared using different approaches were examined to understand the effect of cold rolling and cold spraying on the kinetics of Mg rods. The groups were the (i) Mg-feedstock rods (as-received); (ii) cold rolled Mg-strips obtained after 300 passes; (iii) reactive ball milled Mg/5.5 wt.% Ni obtained after high energy ball milling under H_2_ for 50 h, cold rolled Mg-rods processed for 300 passes and then cold sprayed Mg-strips coated with Ni powders at 150 °C (iv) 1-time, (v) 2-times, and (vi) 3 times.

## 3. Results

### 3.1. Structural Analysis

The XRD pattern of the raw Mg-rods is presented in Figure 5a. Obviously, the sample exhibited pronounced Bragg peaks diffractions related to hcp-Mg (PDF# 00-004-0770). The FE-HRTEM image of the cross-sectional view of this sample showed a Moiré fringe pattern related to hcp-Mg (002), without any evidence of any types of lattice imperfections, such as stacking faults and twins (Figure 6a). This finding was also confirmed by the bright field (BF) TEM analysis of the cross-section for the as-received Mg-rods, which revealed perfect grain boundaries with no indication of the presence of lattice imperfections (Figure 6a). Moreover, the as-received Mg-rods composited of large grains (182 nm to 578 nm), as shown in Figure 7a. Introducing severe lattice imperfections upon rolling for 10 times led to the formation of dense dislocation walls and a larger number of grain boundaries (Figure 7b). In addition, those large Mg grains shown in Figure 7a were divided along their boundaries into smaller elongated grains with sizes ranging from 52 to 468 nm, as shown in Figure 7b.

The formation of microscaled intimated bands formed as a result of shear stresses applied to the cold-rolled Mg-rods can be seen in the FE-SEM micrograph for the sample obtained after 100 cold rolling passes (Figure 7c). The XRD analysis confirmed a significant shift of the two major hcp-Mg Bragg peaks, (100) and (002) to the high-angle side, where the major crystallographic plane (101) had completely disappeared for the sample obtained after 300 passes of cold rolling (Figure 5b). This result indicated a significant lattice imperfection during the CR phase [27,28,29,52,53] as well as the development of a very high degree of fiber texture in the plane (002) [52]. The internal strain indicated by the broadening of Bragg peaks for the sample obtained after 300 passes of CR (Figure 6b) is attributed to the presence of a high dislocation density and grain refinement. The intensity of crystallographic plane (002) was connected to a twinning volume fraction originating from crystal lattice reorientation during twinning, as shown in Figure 6. Towards the end of cold rolling process (300 passes), the sample had twin nuclei that overlapped with dissociated dislocation and stacking faults from the grain boundary, as presented in Figure 6b.

An explanation of this finding is that the deformation that developed in hcp-metals (e.g., Mg, Ti, and Zr) is typically accommodated by twinning and basal slip deformation mechanisms [54]. Furthermore, as shown in Figure 6b, the cold-rolled sample had various forms of twins, including V-shaped and deformation twins. As a result, and based on Jorge et al. interpretations [28] the twinning occurred during the cold rolling phase is responsible for the creation of different amounts of (002) texture. 

The cold-rolling process for 300 passes provided a substantial contribution to the grain refinement of Mg-rods and Zr-rods [27,52,54], as indicated in the BF image taken by scanning transmission electron microscope (STEM) and displayed in Figure 7d. After this final stage, the sample, which exhibited a fine microstructure, consisted of nanoscale grains, ranging from 11 to 28 nm, as displayed in Figure 7d.

### 3.2. Cold Spraying of Mg-Rods with Ni Particles

#### 3.2.1. Morphological Characteristics of Cold-Sprayed Ni-Powder Particles

Figure 8a,c,e display schematics of a supersonicated single Ni particle penetrating the surface of the Mg-strip upon cold spraying with Ni powders. The corresponding FE-SEM micrograph for each illustration is displayed in Figure 8b,d,f, respectively. The aggregated Ni powders used in the present study exhibited an almost spherical morphology (Figure 8a,b) and an apparent particle size of ~3 μm in diameter, as illustrated in Figure 8b. 

These powders, which were composed of ultrafine Ni particles (<400 nm in diameter), tended to agglomerate due to semi-van der Waals interactions between the fine particles [55]. The super sonicated fine Ni-particles pelted into the Mg-substrate were plastically deformed (Figure 8c–e) due to the application of high impact forces generated by the supersonic -DeLaval Nozzle gun (Figure 3b). They embedded into the surface of the Mg-substrate (Figure 8d) to create catalytic sites, as presented in Figure 8d,f. Moreover, Mg-surfaces were substantially affected by those pelted Ni particles, as implied by the stacking faults and nano twinges revealed in the Mg-substrate in Figure 9. These Ni powder particles were continuously deformed and divided into sub-particles, as shown in Figure 8e.

In the present work, cold spray process were applied to coat the Mg-substrate with one, two, and three Ni-layers. The scanning electron image and EDS-maps for Mg and Ni elements of three individual samples coated for one, two and three times are presented together in Figure 10. The first Mg sample coated with Ni for one-time (Figure 10a,b) and displayed heterogeneous distribution of Ni powders, as indicated in Figure 10c. Most of the Mg-surface was not doped, as shown in Figure 10b. An increase in the number of Ni layers (2 coatings) created thicker Ni-layers (Figure 10d), and most of the area of the Mg substrate (Figure 10e) was covered by Ni powders (Figure 10f). The elevation FE-SEM view of the Mg sample coated with three Ni-layers is displayed in Figure 10g. The Mg-surface (Figure 10h) was homogeneously doped with cold sprayed Ni powder particles, as presented in Figure 10i.

Based on the morphological analysis, that when the Ni particles pelted into Mg-strips through a high-velocity jet, they undergo substantial localized plastic deformation together with the Mg-substrate upon the application of impact forces. Thus, employing a supersonic velocity generates impact stress on the particle that is greater than its yield stress [56]. When the impact stresses are applied, high plastic strain rates are achieved in the contact particle/substrate zone within a very short time [57]. This finding is indicated by the stacking faults overlapping with nanotwins in the HRTEM image of the Mg-strip coated three-times with Ni particles (Figure 9). Grujicic et al. estimated that 90% of the applied impact energy on the particle/substrate materials is converted into local heat [58]. Due to the application of such adiabatic heating, local softening of the particles (Ni) and substrate materials (Mg) developed.

One advantage of using cold spraying technique for doping Mg-strips with Ni powder particles, the capability of this process to remove the MgO layer formed on Mg-substrate and create Mg-fresh surfaces. Thus, bonding between the two metallic species (particles and substrate) is successfully achieved, as reported by Hussain et al. [59]. In contrast to the other coating techniques (e.g., sputtering, chemical vapor deposition, and physical vapor deposition), the formed Ni coat contained cavities and pores, as indicated by the gap shown between Ni-particles and the Mg-substrate (Figure 8e,f). These micro-/nano-pores functioned as a hydrogen diffusion gateway, improving the absorption/desorption kinetics of Mg.

#### 3.2.2. DSC Analysis

HP-DSC and He-atmospheric pressure DSC techniques were used to investigate the hydrogenation and dehydrogenation behavior of cold rolled Mg-strips catalyzed with cold sprayed Ni powders (Figure 11). The measurements were conducted for three different samples of the raw Mg-rods, cold rolled Mg-rods after 300 passes, and Mg-rods cold rolled for 300 passes and then cold sprayed three times with Ni powder particles. The HP-DSC curve of the as-received Mg rods obtained at 10 °C/min in a temperature range between 50 and 525 °C under 30 bar of hydrogen gas pressure indicated, the absence of any exothermic hydrogenation/dehydrogenation reactions, as shown in Figure 11a.

These results implied the inability of the large-grain Mg rod to react with reactive hydrogen. When Mg rods were mechanically treated with cold rolling for 300 passes, the size of the Mg grains was dramatically decreased and severely deformed, as shown in Figure 7d. Cold rolling, which induced high-density imperfections, disintegrated submicron Mg grains into nanosized grains (Figure 7d), led to decrease the hydrogen diffusion gap. All of these changes increased the ability of Mg strips to react with hydrogen simultaneously, as characterized by the appearance of an exothermic reaction peak centralized at 285 °C (Figure 11a). The corresponding dehydrogenation reaction was detected by the endothermic reaction event centralized at 470 °C, as displayed in Figure 11a. Both events were confirmed by XRD analysis, conducted for the two samples obtained after the hydrogenation (first peak) and dehydrogenation (second peak) reactions.

The hydrogenation temperature observed for the cold rolled Mg-strip drawn for 300 passes and then cold sprayed three times with Ni powder particles was dramatically decreased to 144 °C, as presented in Figure 11a. This finding may suggest the role of supersonic Ni powders as a catalytic agent to reduce the gas up take reaction. In addition, this obvious improvement in hydrogenation was attributed to introducing a high-density imperfection network to the Mg-strips during the cold spray process, as shown in Figure 8 and Figure 9. The corresponding dehydrogenation reaction measured under 30 bar of H_2_ showed a very limited reduction in peak temperature (463 °C) compared with the cold rolled sample before the Ni coating (470 °C). This result is attributed to the high pressure of hydrogen (30 bar) used during HP-DSC experiments, which prevented easy gas release.

The correlation between the hydrogenation temperature and number of Ni-layers coated the cold rolled Mg-strips was determined from the HP-DSC curves presented in Figure 11b. The Mg-strip cold sprayed with Ni powder one time (1.48 wt.% Ni) successfully reacted with H_2_ at 219 °C, as characterized by the broad exothermic peak shown in Figure 11b. An increase in the number of Ni powder-coatings to two (3.93 wt.% Ni) increased the molar fraction of Ni and improved the hydrogenation behavior of Mg-strips, as characterized by a temperature decrease in the gas uptake reaction to (159 °C), as shown in Figure 11b. The sample coated by supersonicated Ni-powders three times (5.28 wt.% Ni) possessed a high capability to react with hydrogen at a relatively low temperature (144 °C), as presented in Figure 11b.

The HP-DSC measurement recorded, using k of 10, 11, and 12 °C/min enabled us to calculate the apparent activation energy (E_a_) of hydrogenation at 30 bar of H_2_. The HP-DSC thermograms of cold rolled Mg-strips cold sprayed with Ni-powders three times examined at different k values are displayed in Figure 12a. All samples obtained with different k values revealed exothermic reactions at different peak temperatures (Figure 12a) that were related to the formation of the MgH_2_ phase. The E_a_ of hydrogenation was obtained using Arrhenius equation;
ln k = ln A − Ea/RT(1)
E_a_ = −RTln(k)(2)
where k is a temperature-dependent reaction rate constant, A is the Arrhenius parameter, E_a_ is the activation energy, R is the gas constant, and T is the absolute peak temperature of hydrogenation. The E_a_ of hydrogenation was determined by measuring the hydrogenation peak temperature obtained at different k values and then plotting ln(k) versus 1/Tp, as shown in Figure 12b. The best fit for the results calculated using the least-square method indicated that all data points lie closely on the same straight line for an E_a_ of 20 kJ/mol.

Part of the cold rolled Mg-strips doped with Ni powders three times was hydrogenated under 15 bar of hydrogen at 250 °C for 30 min, using a Ti-hydrogen reactor. The DSC curves for these hydrogenated-cold-sprayed samples recorded at 5, 10, 20, 30, and 40 °C/min are shown in Figure 12c. All traces revealed low and high temperatures endothermic events, corresponding to the composition of β- and γ-MgH_2_ phases respectively (Figure 12c). Cold spraying of Mg strips with Ni powders three times (5.28 wt.%) led to a substantial improvement in the decomposition characteristics of the sample and showed a lower E_a_ value (74 kJ/mole). In parallel to the catalytic effect of Ni, the improvement observed in the decomposition process was due to the texture and the presence of defects [58].

#### 3.2.3. Hydrogenation/Dehydrogenation Kinetics

Six separate batches of Mg samples collected using different catalyzation methods with Ni powders, were analyzed under the same measurement conditions. The batches were (i) as-received Mg feedstock rods; (ii) as-cold rolled Mg-rods, drawn for 300 passes; and (iii) as-reacted ball milled Mg powders doped with 5.5 wt.% Ni under 15 bar of H_2_ for 50 h, as well as cold rolled Mg-rods obtained after 300 passes and then cold sprayed with Ni powders for (iv) 1, (v) 2, and (vi) three times.

The hydrogenation kinetic behaviors for the six batches measured at 150 °C under 10 bar of H_2_, are presented in Figure 13. All the samples revealed a good ability to absorb hydrogen at a different time scales when reacted at a relatively low temperature (150 °C) and moderate hydrogen pressure (10 bar), as displayed in Figure 13. However, the as-received Mg-rods possessed very slow uptake kinetics, as evidenced by the long time (2 min) required to absorb 0.9 wt. % H_2_ (Figure 12). This sample required 144.6 min to completely absorb 5.8 wt.% H_2_, as shown in Figure 13. Surprisingly, the as-doped Mg powders with 5.5 wt.% Ni and reactively ball milled under hydrogen for 50 h did not show a further improvement in hydrogenation kinetics, as indicated by the long time (142.8 min) required to absorb approximately 5.6 wt.% H_2_, which was similar to the feedstock Mg-rods (Figure 13).

The as-cold rolled sample for 300 passes did not demonstrate a noticeable change in hydrogenation kinetics within the first 2 min of the hydrogenation process (1.2 wt.% H_2_). It, however, showed a noticeable rise in hydrogen absorption after 50 min, as shown by its ability to consume 5.1 wt.% H_2_ (Figure 12). 

The hydrogenation kinetics of Mg-strips obtained after cold rolling for 300 passes and cold spraying with various concentrations of Ni powders, on the other hand, showed a notable improvement depending on the amount of Ni coating layers (i.e., Ni-concentrations). After a short hydrogenation time (2 min), the samples that were doped with 1.48, 3.93, and 5.28 wt.% Ni absorbed 4.2, 4.35, and 4.8 wt.% H_2_, respectively (Figure 13). The cold rolled sample, which was coated three times with Ni powders, doping Mg powders with 5.5 wt.% Ni using the reactive ball milling technique did not demonstrate significant hydrogenation kinetics, as clearly presented in Figure 13.

Doping the cold-spray Mg-ribbons with 5.28 wt.% Ni powders through cold spray coating showed a substantial improvement in hydrogenation kinetics (6.1 wt.% H_2_/5.1 min/150 °C/10 bar), as displayed in Figure 13. The ability of the cold spray coating to generate more vacancies and dislocation sites on the surface of Mg-strips (substrate) when pelted by Ni powders at supersonic speeds is credited with this improvement. The Ni powders deposited on the squeezed and plastically deformed Mg-strips on the substrate appeared to insert into the subsurface of the Mg-substrate, encouraging good weldability between Mg and the catalytic agent. We hypothesize that the cold rolling method is more successful than ball milling for increasing hydrogenation kinetics in the reaction of cold rolled Mg-strips. Multiple lattice imperfections and deformations were created on the Mg metal lattice during the cold rolling process, destabilizing the metal and increasing its ability to absorb hydrogen. However, a second phase for catalyzing the cold rolled Mg strips with a proper catalytic agent(s), such as Ni powders and/or any other catalytic agent, should be included in this procedure (s).

The corresponding dehydrogenation kinetics of the six batches of Mg strips are shown in Figure 14. All of the results were obtained at 200 °C with a hydrogen pressure of 200 mbar. As shown in Figure 14, all of the samples were able to release their stored hydrogen, with the exception of the as-received Mg-rods, which failed to perform a dehydrogenation reaction even after 301 min. The as-doped Mg strips with 5.5 wt.% Ni powders that were reactively ball milled under hydrogen gas for 50 h showed only a minor improvement in dehydrogenation kinetics, as demonstrated by the very long period (301 min) taken to release approximately 5 wt.% H_2_ (Figure 14). The sample prepared by cold rolling for 300 times did not show a noticeable improvement in dehydrogenation kinetics during the first 6 min of the dehydrogenation method (−0.3 wt.% H_2_). However, after a very long period (90.1 min), it displayed an apparent rise in hydrogen release, as shown by its ability to desorb −4.7 wt.% H_2_ (Figure 14).

As shown in Figure 14, Mg-strips obtained by cold rolling for 300 passes and then cold spraying with different concentrations of Ni powders demonstrated excellent dehydrogenation kinetics. Compared to the single-coated sample, the Ni two- and three-coated samples desorbed −0.8 and −2.3 wt.% H_2_, respectively, after just 6 min, as shown in Figure 14. The Mg-strips were cold rolled for 300 passes before being cold sprayed with Ni once and twice, resulting in −4.7 wt.% H_2_/153.5 min and −5.9 wt.% H_2_/52.9 min, respectively. In comparison, after just 21.1 min, the sample cold sprayed with Ni three times was saturated at −6.1 wt.% H_2_, as shown in Figure 14.

Figure 15a shows the XRD patterns of cold rolled Mg-strips that were cold sprayed with Ni 3 times recorded after the hydrogenation process was completed. The sample revealed strong Bragg peaks associated with the MgH_2_ process, which coexisted with a small volume fraction of unreacted Mg (Figure 15a). The Ni coating material is depicted by the very large Bragg lines in the figure. During the hydrogenation process depicted in the figure, a small volume fraction of the Mg_2_NiH_4_ phase was generated by the reaction between Mg and Ni. It is assumed that the presence of this reacted phase enhances the dehydrogenation kinetic.

Figure 15b shows the XRD pattern of the Mg strip coated three times with Ni after the dehydrogenation process was completed at 200 °C/200 mbar of hydrogen for 5.1 min (b). As shown in Figure 15b, the sample had prominent Bragg peaks due to the coexistence of the hcp-Mg process with a minor volume fraction of undecomposed-MgH_2_ (b).

The cycle-life-time of Mg-strips that were cold rolled for 300 passes and then cold sprayed with Ni three times is displayed in Figure 16. This system shows good hydrogenation properties, indexed by long cyclic stability even after about 350 h with minimal degradation on the hydrogen storage capacity, which exhibited nearly constant absorption and desorption values of 5.2 wt.% H_2_, as displayed in Figure 16.

## 4. Discussion

Apart from the traditional ways of hydrogen storage, Mg and Mg-based materials have been considering as the most candidate hydrogen storage media for real applications. The worldwide interest on Mg metal is attributed to its natural abundance, light weight, and its capability to store hydrogen up to 7.60 wt.%. In spite of these attractive properties of MgH_2_, MgH_2_ in its pure form has a high stability and shows very slow kinetics of hydrogenation dehydrogenation at temperatures less than 300 °C. Within the last two decades, enormous efforts have been dedicated in order to improve the hydrogenation/dehydrogenation behaviors of MgH_2_ throughout mechanical treatment regime and/or doping the hydride phase with proper catalytic agents. It is a good practice to combine these two strategies, which led to excellent improvement in the hydrogen storage behavior of MgH_2_ [52]. 

In contrast to the catalyzation of MgH_2_ by doping with metallic powders via ball milling method, the present study tried to introduce a new catalyzation process with different concentrations of Ni powders, using cold gas dynamic spraying technique. In this process, a supersonic stream of solid powders is pelted toward a target substrate, where they penetrate the substrate’s surface through a large number of pores, resulting in a coat of the desired thickness. According to the morphological examinations, when the Ni particles (catalytic agent) are pelted into Mg strips through a high-velocity jet and impact forces are applied, they undergo extreme localized plastic deformation along with the Mg substrate. When a supersonic velocity is applied, the impact stress on the particles (Ni) is greater than the yield stress. High plastic strain rates in the contact particle/substrate region are reached in a very short period when impact stresses are applied. The development of stacking faults overlapped with nanotwins in the Mg strips, as seen in the HRTEM image of the Mg strip sample obtained after the application of three coatings of Ni particles, supports this conclusion. Cold spraying shows an excellent capability to break the MgO layer formed on the Mg substrate and create fresh Mg surfaces upon pelting with Ni “bullets”. Thus, bonding between the two metallic species (Ni particles and Mg substrate) was successfully achieved in the absence of an oxide layer. In contrast to the other coating techniques (e.g., sputtering, chemical vapor deposition, and physical vapor deposition), the Ni coat formed by cold spraying contained cavities and pores, as indicated by the gaps observed between Ni particles and the Mg substrate. These micro-/nanopores function as a hydrogen diffusion gateway, improving the absorption/desorption kinetics of Mg. Moreover, the cold spraying technique is expected to be industrially used as a cost-effective catalyzation process to prepare bulk/sheets/foil/disks of Mg/MgH_2_ doped with wide selection of catalytic agents, such as metal or metal compounds including intermetallics, alloys, metastable metallic glassy and amorphous alloys, oxide, and carbides, at any desired scale.

Based on the results of the present work, Mg-strips that were cold rolled for 300 passes and then cold sprayed with Ni three times showed a lower E_a_ value (74 kJ/mol). The apparent E_a_ of our system is less than those values for reported for pure MgH_2_(124 kJ/mol) [24] Mg_85_In_5_Al_5_Ti_5_ (125.2 kJ/mol) [60], MgH_2_/10 wt.% SrTiO_3_ (109 kJ/mol) [61], MgH_2_/15 wt.% VNbO_5_ (99 kJ/mol) [62], MgH_2_/5 wt.% K_2_NbF_7_ (96.3 kJ/mol) [63], MgH_2_/10 wt.% TiO_2_/supported C (106 kJ/mol) [2], MgH_2_/10 wt.% K_2_NbF_7_/5 wt.% of MWCNT (~76 kJ/mol) [2], and MgH_2_/30%Ni (83 k/mol) [64] systems. In contrast, it is higher than, MgH_2_ mixed with Ti_0.4_Cr_0.15_Mn_0.15_V_0.3_ powders (71.2 kJ/mol) [2], and V (64.7 kJ/mol) [35]. 

## 5. Conclusions

A new catalyzation process, using cold spray technique was employed to improve cold-rolled Mg-strips. This technique was used to boost the hydrogen storage properties of cold-rolled Mg strips obtained after 300 passes. Ni powder particles, which acted as the catalytic agent, were pelted toward the Mg strips using a supersonic jet at a speed of 500 m/s at 150 °C under a high argon gas pressure. Based on the results of the present study, pelting the Mg strips with Ni powders produced the desired modification of Mg surfaces. In this coating process, the plastically deformed Ni bullet particles penetrated the surface of the Mg substrate to create micro/nanopores that functioned as a hydrogen diffusion gateway. The decomposition temperature of as-cold rolled sample for 300 passes and then coated with Ni 3 times decreased to 288 °C. The apparent activation energy of dehydrogenation was 74 kJ/mol. This sample possessed attractive hydrogenation/dehydrogenation kinetics at a relatively low temperature (150 °C/200 °C). The sample absorbed/desorbed 6.1 wt.% H_2_ within 5.1/11 min. Moreover, the fabricated system has a high capability to achieve continuous cyclic hydrogenation/dehydrogenation processes of 350 cycles at hydrogenation/dehydrogenation temperature of 250 200/225 °C. No severe degradation in the hydrogen storage capacity could be detected even after 350 h of continuous cycles. Moreover, the kinetic of hydrogenation/dehydrogenation processes remaining nearly constant without failure or serious decay. 

## 6. Patents

M. Sherif El-Eskandarany, Mohammad Banyan and Fahad Al-Ajmi. Method for Doping Magnesium with Nickel by Cold Spray Technique. US 10,443,132 B1, 30 October 2019.

## Figures and Tables

**Figure 1 nanomaterials-11-01169-f001:**
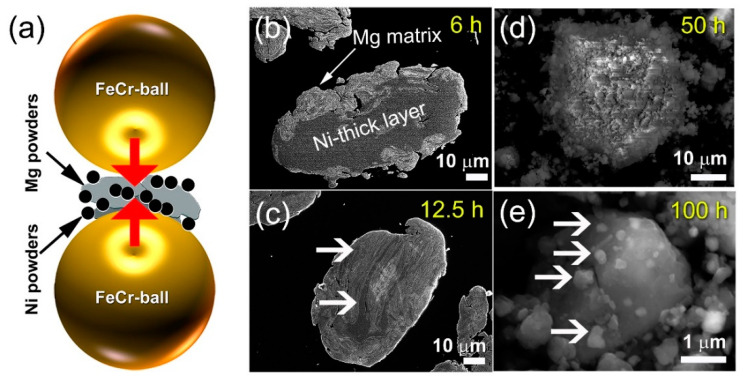
(**a**) During hydrogen-RBM of Mg doped with 5.5 wt.% Ni powders, a ball-powder-ball collision occurs. In (**b**) and (**c**), FE-SEM micrographs of the cross-sectional view of the powders obtained after 6 h and 12.5 milling, respectively, are shown. The FE-SEM micrographs of the powders milled for 50 and 100 h, respectively, are shown in (**d**,**e**), respectively.

**Figure 2 nanomaterials-11-01169-f002:**
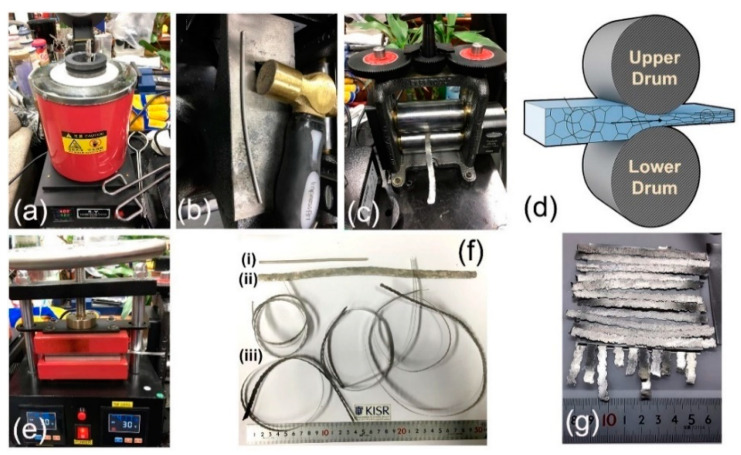
(**a**) Induction melting furnace hosted the Mg-rods, (**b**) forging process, (**c**) cold rolling process, (**d**) schematic illustration of drawing a Mg bar, using two-drum cold roller machine, (**e**) warm pressing, (**f**) original Mg-rod (**i**), Mg-strip obtained after forging at 400 °C for 200 times (**ii**), (**iii**) final product after re-cold rolling for 10 passes. Final product of Mg ribbons obtained after 300 passes of cold rolling, and (**g**) array of snipped Mg-strips ready for coating with Ni powders, using cold spray technique.

**Figure 3 nanomaterials-11-01169-f003:**
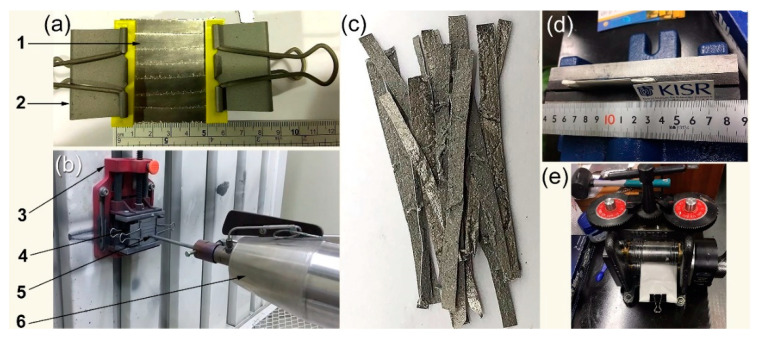
(**a**) The cold-rolled Mg ribbons (1) were placed and fixed on stainless steel plate by clips (2); (**b**) two-movable jaws (3) were used to fix the ribbons that was aligned perpendicular (5) to the cold spray (CS) gun nozzle (6). In the image above (**c**), the Mg-ribbons have been coated three times with Ni powders. To ensure straightening, the uncoated Mg-strips were wrapped in balance papers and put in a two-jaw style vise for 16 h (**d**), and then CR for 10 times (**e**).

**Figure 4 nanomaterials-11-01169-f004:**
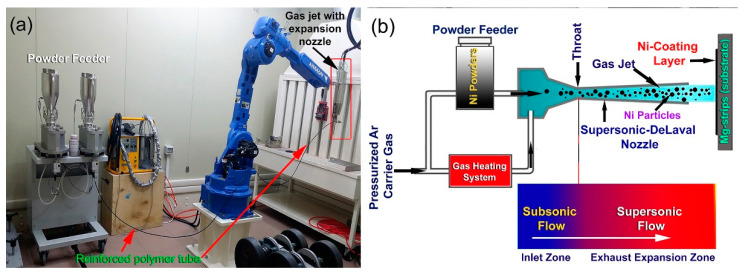
(**a**) Cold-spray experimental set up, and (**b**) schematic presentation of the coating CS process of Mg-ribbons, using supersonic Ni powder particles. The subsonic and supersonic zones within the nozzle gun are depicted in the figure (**b**).

**Figure 5 nanomaterials-11-01169-f005:**
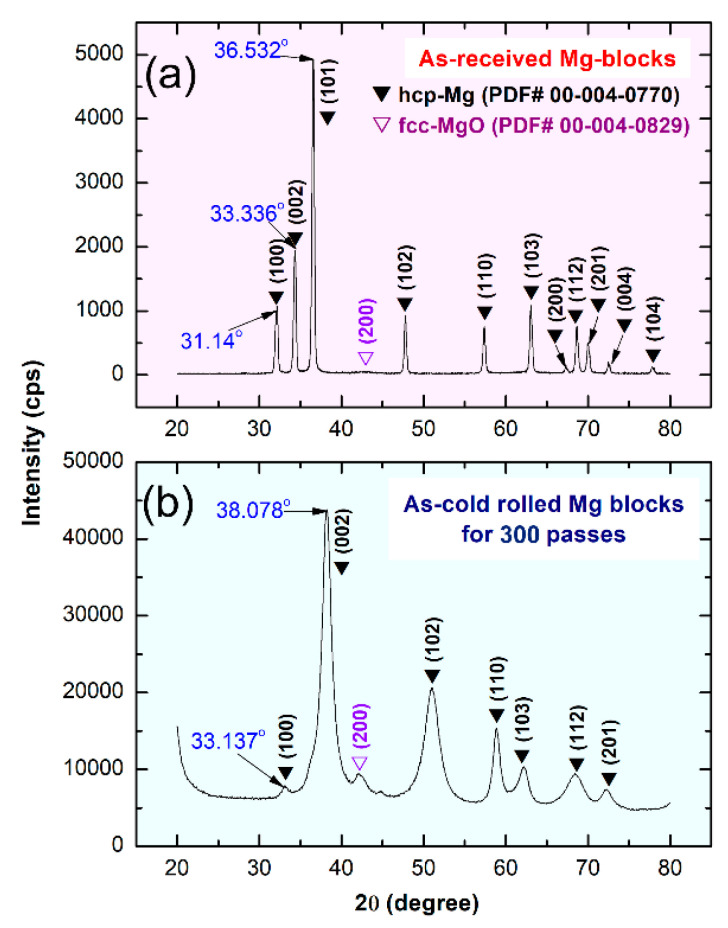
XRD patterns of the raw Mg-rods before and after cold rolling are shown in (**a**,**b**), respectively.

**Figure 6 nanomaterials-11-01169-f006:**
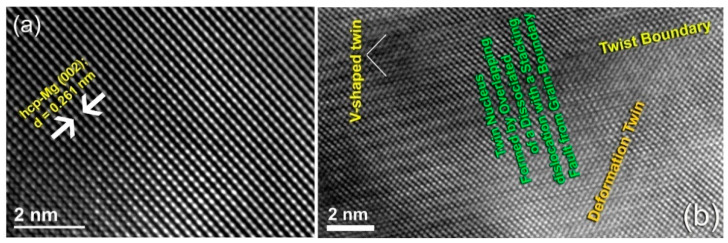
FE-HRTEM image of (**a**) the starting Mg-rods and (**b**) the corresponding FE-HRTEM image of Mg-rods drawn for 300 passes of cold rolling.

**Figure 7 nanomaterials-11-01169-f007:**
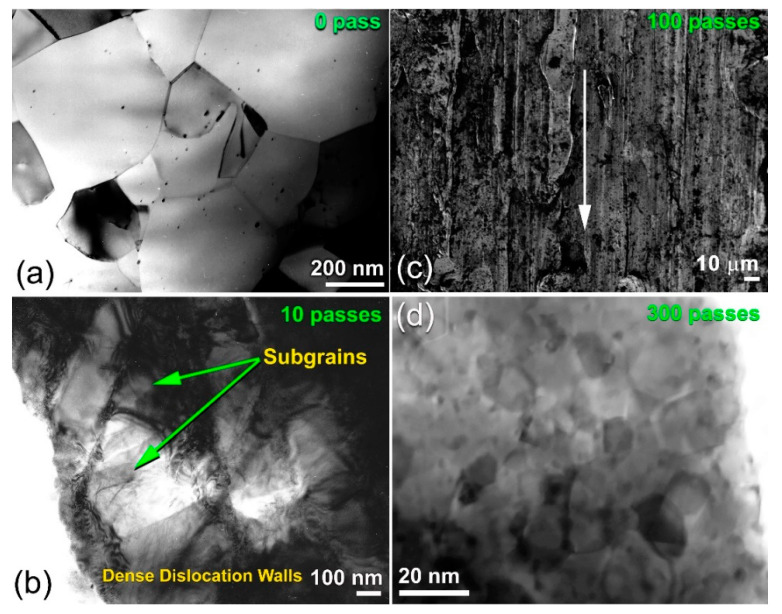
Low magnification BFI of raw Mg-rods obtained after cold rolling for (**a**) 0 passes and (**b**) 10 passes. The low magnification FE-SEM micrograph of Mg-rods obtained after 100 passes is displayed in (**c**), where the STEM of the rods drawn for 300 passes of CR is shown in (**d**).

**Figure 8 nanomaterials-11-01169-f008:**
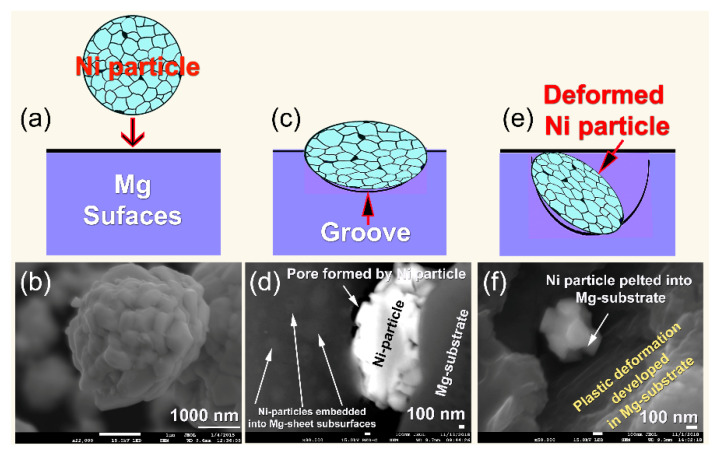
Schematics (**a**,**c**,**e**) and FE-SEM micrographs of Ni powders (**b**,**d**,**f**) pelted into Mg-substrate strips.

**Figure 9 nanomaterials-11-01169-f009:**
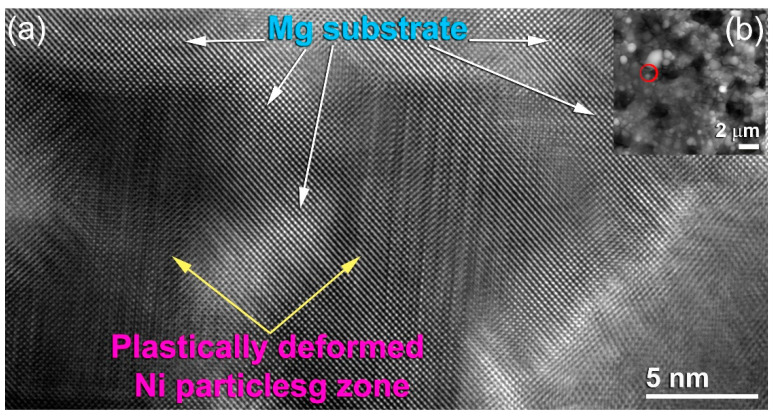
FE-HRTEM with atomic resolution of a selected zone of Mg-strip coated with 3 Ni powder layers, using the cold spray process. The FE-HRTEM with atomic resolution of a selected zone of Mg-strip coated with 3 Ni powder layers, using the cold spray process is presented in (**a**). The corresponding image of scanning transmission electron microscope (STEM) is presented in (**b**).

**Figure 10 nanomaterials-11-01169-f010:**
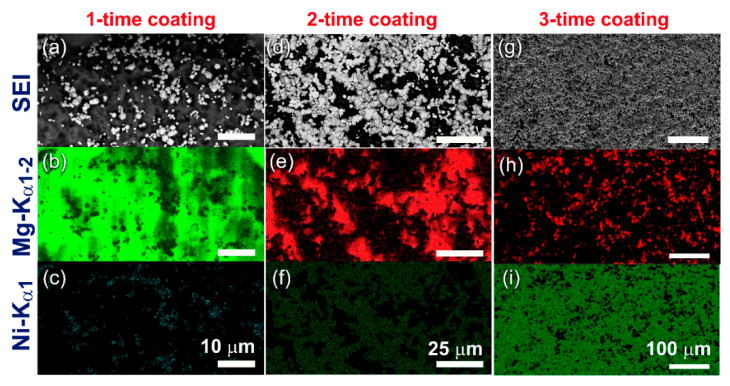
Scanning electron images (SEIs) of Mg-strips coated with Ni powder particles (**a**) 1-time, (**d**) 2-times, and (**g**) 3-times. The corresponding EDS elemental maps for Mg in the samples coated 1-, 2-, and 3-times are presented in (**b**), (**e**) and (**h**), respectively. The Ni-EDS maps of the CS samples obtained after 1-, 2-, and 3 coatings are presented in (**c**), (**f**), and (**i**), respectively.

**Figure 11 nanomaterials-11-01169-f011:**
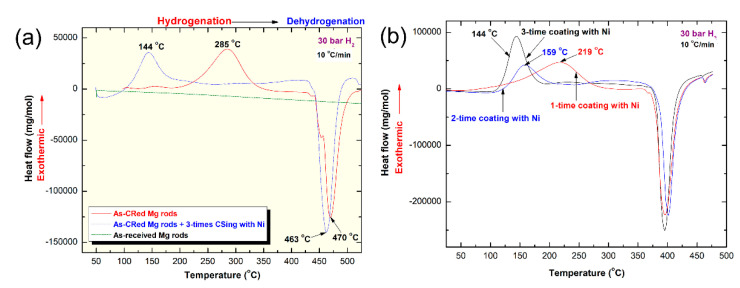
HP-DSC thermograms of (**a**) the as-received Mg rods, as-cold rolled rods, as-cold rolled that were cold sprayed with Ni powders 3-times and (**b**) Mg-rods that were cold rolled 300 times and then cold sprayed with Ni powders 1-, 2-, and 3-times. The HP-DSC thermograms measured with different heating rates, k (10, 11, and 12 °C/min) of cold rolled Mg-rods for 300 times and then cold sprayed with Ni powders for 3 times are displayed in (**c**) together with Arrhenius plot of hydrogenation (**d**). The He-atmospheric pressure DSC thermograms measured with different k values (5, 10, 20, 30, and 40 °C/min) of cold Mg-rods that were cold rolled 300 times and then cold sprayed with Ni powders for 3 times are presented in (**e**).

**Figure 12 nanomaterials-11-01169-f012:**
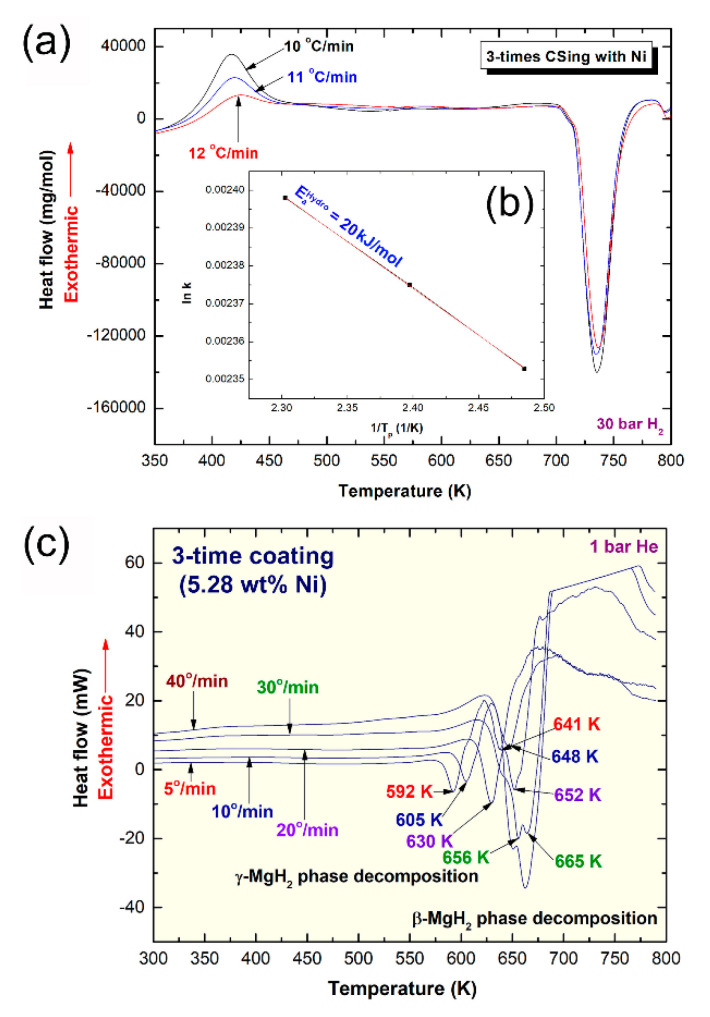
The HP-DSC thermograms measured with different heating rates, k (10, 11, and 12 °C/min) of cold rolled Mg-rods for 300 times and then cold sprayed by Ni powders for 3 times are displayed in (**a**) together with Arrhenius plot of hydrogenation (**b**). The He-atmospheric pressure DSC thermograms measured with different k (5, 10, 20, 30, and 40 °C/min) of cold rolled Mg-rods for 300 times and then cold sprayed with Ni powders for 3 times are presented in (**c**).

**Figure 13 nanomaterials-11-01169-f013:**
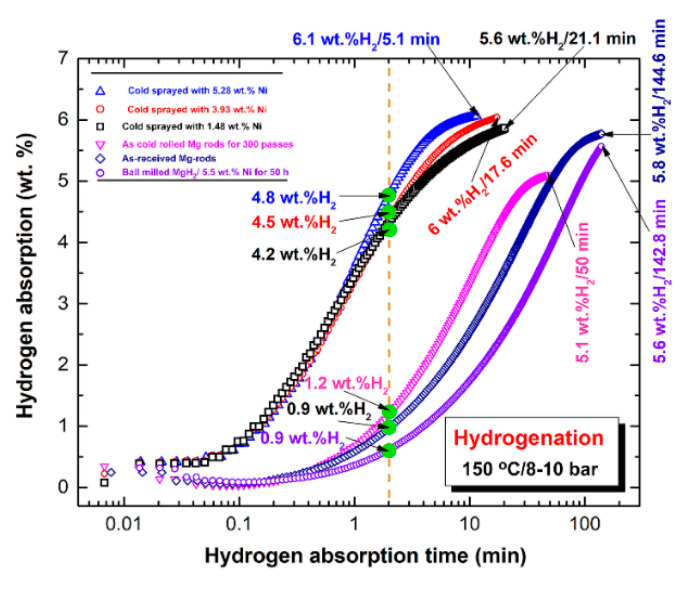
Hydrogenation kinetics of the as-received Mg-rods, Mg-rods that were cold rolled for 300 passes, Mg-rods that were cold rolled for 300 passes and then cold sprayed with Ni powders 1-, 2-, 3 times; and ball-milled MgH_2_/5.5 wt.% Ni obtained after 50 h of milling. The measurements were conducted at 150 °C/10 bar.

**Figure 14 nanomaterials-11-01169-f014:**
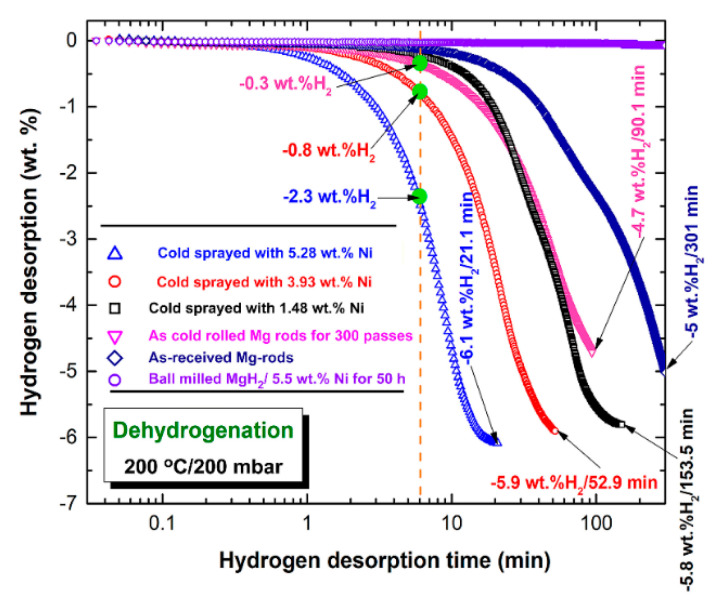
Hydrogen released kinetics of raw Mg-rods; Mg-rods that were cold rolled for 300 passes, Mg-rods that were cold rolled for 300 passes and then cold sprayed with Ni powders 1-, 2-, 3 times; and ball-milled MgH_2_/5.5 wt.% Ni obtained after 50 h of milling. The measurements were conducted at 200 °C/200 mbar.

**Figure 15 nanomaterials-11-01169-f015:**
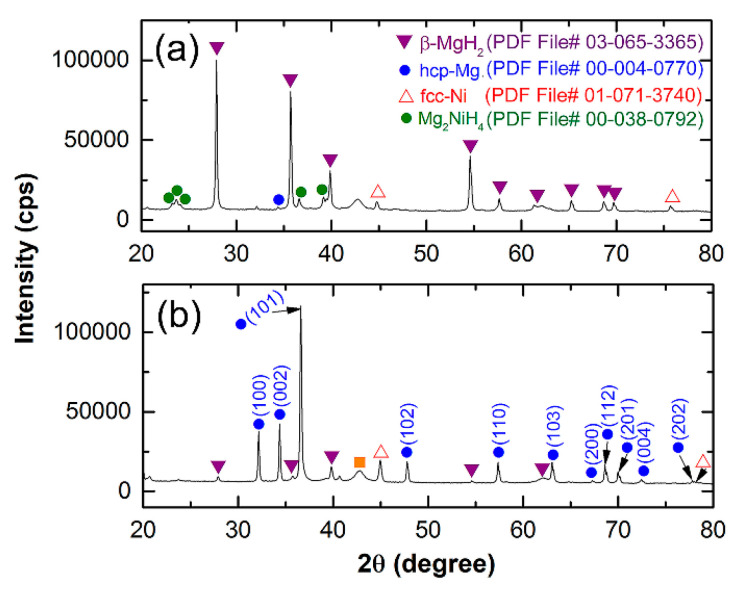
XRD patterns of Mg-strips that were cold rolled for 300 passes and then cold sprayed with Ni 3 times after completion of the (**a**) hydrogenation and (**b**) dehydrogenation measurements.

**Figure 16 nanomaterials-11-01169-f016:**
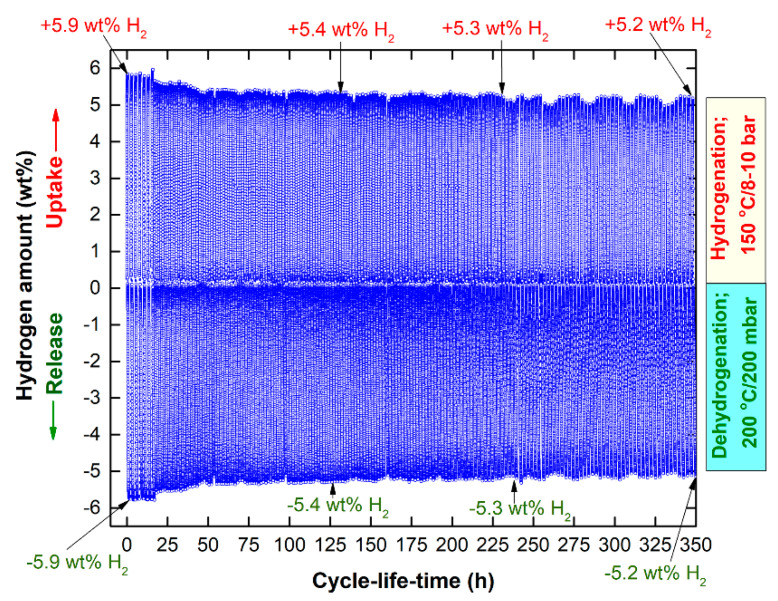
Cycle-life-time of Mg-strips that were cold rolled for 300 passes and then cold sprayed with Ni 3-times.

**Table 1 nanomaterials-11-01169-t001:** Hydrogen Storage Technology Comparison *.

	Compressed-H_2_	Liquid-H_2_	MgH_2_
Pressure (MPa)	70	1	1
Gravimetric Energy Density (wt.%)	5.7	7.5	7.6
Volumetric Energy Density (MJ/L)	4.9	6.4	13.2
Temperature (°C)	Ambient	−253	300

* Reference [8].

## Data Availability

Not available.

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
