# Peer review of "Cold Gas-Dynamic Spray for Catalyzation of Plastically Deformed Mg-Strips with Ni Powder"

_nanomaterials, 2021, doi:10.3390/nano11051169_

Round 1
Reviewer 1 Report
This work is about the improvement of mainly the hydrogen kinetic behavior of magnesium hydrides by adding Ni particles through cold spraying. Mg rods are first cold rolled and then cold-rolled Mg ribbons are cold spraying. Some improvement in the kinetic behavior has been observed.
Abstract
The abstract should be shortened.
Introduction
- Page 2: “In practical 46 terms, hydrogen may be stored either in the gaseous state under 350 – 700 bar or as liquid 47 phases at -253 o These traditional approaches, on the other hand, are costly and mismatching with the safety requirements [6].”
This statement is partially true. It is advisable to support this statement with numbers. It is possible to say that in terms of security, metal hydrides are extremely pyrophoric and the cost of the material would be an issue for large-scale applications. Therefore, I would suggest pointing out the advantages of the solid storage hydrogen method in comparison to the traditional physical methods, which has been technologically accepted (liquid hydrogen in aerospace applications) or/and commercially accepted (high-pressure hydrogen reservoirs), instead of strongly write that a mismatch between the traditional approaches and the requirements exist, since it is partially true.
- In section 1.2, it is true that Mg has some good capabilities as hydrogen storage materials, however, the range is quite large, i.e. large scale, small scale, mobile or stationary applications. At this point, there are works about the use of Mg as storage material and the main constraint of Mg at a large scale is the heat management given by its high reaction enthalpy. Therefore, section 1.2 should be improved providing the readers a better overview of the system Mg/MgH2 as hydrogen storage material for a system.
- In the section drawbacks of the catalyzation, on my view, it is not the catalyzation that is not working well, according to the “results” shown in Fig. 1. It is the ball milling method. It is well known that by milling a ductile material like Mg with a relatively brittle material as Ni, the Mg acts as a matrix for Ni and it might happen that the distribution is not as uniform, which is questionable. Therefore, I suggest changing the name of this subsection to a drawback of high-energy ball milling for homogeneous distribution of catalyst or something of the kind.
Results
- The analysis of the activation energy should be better described. From calorimetric measurements, the method used for the determination of the Ea is the “Kissinger” method, and it is not clear in the paper that this method was applied. It is strongly advisable to revise the procedure used to calculate the Ea. Moreover, at the time to calculate the Ea, please add the error band. A value with two decimals does not make too much sense, since in general, the error bar is larger than the first decimal number.
If you took “k”, the heating ramp, as the k for the Arrhenius equation, the Ea has not been properly calculated. There are several works, in which the addition of 3d transition metals leads to Ea of dehydrogenation ranging between 60-70 kJ/ mol H2 (please check literature)
- In Fig. 12c, why it the peaks between 592-630 K are attributed to metastable phases of Mg (gamma), if there is evidence of the formation of Mg2NiH4 upon hydrogenation (XRD, Fig. 15a). In most of the cases, there is a synergetic effect in which the decomposition of Mg2NiH4 enhances the decomposition of MgH2. There is literature about this. Moreover, the XRD in Fig. 15 should be before presenting the thermal evaluation.
- In the curves presented in Fig. 13, it is notable that the shape of the curve describes a nucleation and growth mechanism for the hydride phase formation. This fact can be confirmed by applying gas-solid models and identifying the rate-limiting step. If the case is so, the effect of the applied cold rolling and cold spray procedure are on the change of the rate limiting step from diffusion (traditional constraint for Mg) to nucleation and growth. I strongly suggest trying to do this analysis, it will enhance the value of the work and provide more inside. The same is valid for the dehydrogenation curve in Fig. 14. However, in this case, traditionally the rate-limiting step for MgH2 decomposition is the movement of the Mg/MgH2
Discussion
- The discussion should be improved. It is supposed to have more scientific background and discuss the results based on the comparison with the published results in order to provide the readers a deeper understanding of the contribution of the work and the mechanisms involved. It is not the case in the discussion.
Conclusion
- Should be re-written according to the suggestions.
References
Please, try to update the references, particularly in the introduction part. Moreover, some titles are not written rightly. Check also all the information in the references.

Author Response
The authors of the present study would like to thank the respected reviewer for his time, knowledge and kind advices. These comments will lead to improve the quality of our work and enriching the discussions. Thank you very much.
We are pleased to bring to his kind attention that all of his comments and suggested changes have been seriously considered in this revised version.
Abstract
The abstract was shortened.
Introduction
- Thank you very much for the suggested and vital remarks. We have followed these suggestions and provided some comparisons between the three hydrogen storage technologies with supported references that enriched section 1-2.
- We do so much agree with you on changing the sub-title into “Drawbacks of high-energy ball milling”. Thank you very much.
Results
- We would like to thank you very much for the fruitful instructions and suggestions. Unluckily, we had no enough machine time on using the high pressure device to make another 2 or three runs. Thus, the calculations were based on a single hydrogenation run. The details of the DSC experiments are highlighted (page 13). Thank you very much.
- Yes, there is a synergetic effect upon the formation of Mg2NiH4 In fact, the XRD patterns shown in Fig. 15 was for the sample taken after measuring the dehydrogenation kinetics.
- We do so much apologize for not be able at present to do such an important analysis. We are keeping this analysis in mind for a different work in the near future.
Discussion
This section was revised based on the comments and suggestions made by the respected reviewer.
Conclusions
The conclusion was revised and modified.
References
The references were updated.
Again, thank you very much
Reviewer 2 Report
This paper reports the cold spray technique was used to boost the hydrogen storage properties of cold-rolled Mg strips obtained after 300 passes. This topic is interesting and may be helpful for the increase of the application of the Mg-based hydrogen storage materials in the future. This manuscript is suitable to be published in Nanomaterials after addressing the following points.
- The catalytic effects of transition metal/element for hydrogen storage properties of hydrides, such as MgH2 and LiBH4 have been verified and proposed many times. In this manuscript, the fabricated nanocomposite MgH2/5.28 wt% Ni strips have shown the capability to desorb 6.1 wt% of hydrogen gas within 11 min at 200 oC under 200 mbar of hydrogen pressure, which is very impressing. However, I am wondering how is cycling stability of the MgH2/5.28 wt% Ni system, this is one of the most important properties of hydrogen storage materials, please provide more details.
- The Introduction part should be more concise.
- How to avoid the agglomeration of the sprayed Ni particles? How to control the amount of the Ni particles since I don’t think all the particles can be coated on the Mg surface successfully.
- Dual-tuning effects of the thermodynamics and kinetics for hydrogen storage materials is the real key issues for hydrogen economy and hydrogen storage materials, especially for the metal hydrides such as MgH2. The newly developed ball milling with aerosol spraying (BMAS) for dual-tuning the thermodynamics and kinetics of MgH2+LiBH4 system should be paid attentions and may compare the results with some closely related reports on multiple hydride systems.
Author Response
- Firstly, we would like to thank you very much for the critical reading of the manuscript and for the serious suggestions and comments. Your kind encouragement words were highly appreciated.
- We do so much agree with you that examining the cyclability of any hydride system, particularly MgH2-system is one of the crucial characteristics that should be tackled in part to consider the system for real applications. In fact, we have not added Fig. 16 (Cycle-life-time) to the manuscript in order to avoid the limitations of page number. Based on you kind and vital suggest, this figure was included and discussed in the revised version.
- In this regard, to the cold spray processing and its parameters used to catalyze Mg-strips with Ni powders, we are pleased to answer your kind questions as following;
- The Ni-particles catalytic agents were sonicated with little amount of ethanol for 30 min prior to charging them in the feeder. The suspension was then dried at 200 oC for 6 h to evaporate the ethanol.
- This simple sonication method succussed to overcome the agglomeration effect of the Ni-powder during the cold spraying process, however, some of these fine particles prefer to form large aggregate during the process due to the Van der Walls effect.
- More importantly, in this catalyzation approach, the Ni particles were carried by a stream of Ar gas via a high-velocity jet at a supersonic velocity. Accordingly, the pelted Ni particles penetrated the Mg -substrate ribbons and hence created numerous micropores into the Mg. This allowed the Ni particles to form a homogeneous network of catalytic active sites in Mg substrate.
- Furthermore, both of the Ni-catalytic agent and Mg matrix were experiences from severe plastic deformation that led to enhance the kinetics of the hydrogenation/dehydrogenation processes.
- In fact, we still need to tackle more detailed experiments with another catalytic agents of metal and alloys to ensure the validity of the process.
- After finishing coating with ten lines of Ni, the weight of Mg-substrate balanced before the process, was balanced again to get the weight of Ni. This process was repeated 5 times to take the average value.
- Based on these results, we have realized the amount of Ni of each coating time.
- The exact Ni concentrations, we then examined by SEM/EDS technique. Before starting the real catalyzation experiments, we have calibrated the machine with fixed powder flow rate, and fixed gun speed to make the number of the desired layers.
- We do so much agree with you that BMAS, taking MgH2-LiBH4 system as a typical example would be find wide range of applications as powerful technique leads to enhance the hydrogen storage properties of metal hydride systems. We are planning to use this technique in the near future for doping both of MgH2 powders and plastically deformed Mg-strips.
Again, thank you very much.
Reviewer 3 Report
In this paper, the authors investigated the possibility of utilizing a cold spraying (CS) technique to catalyze the Mg-substrates ribbons with Ni powders. The results of this paper are very good presented and discussed in terms of crystal structure, morphology, kinetics, and thermal stability. The paper is well written. The methods are very good presented.
Base on these, it may suitable to publish this manuscript in Coatings, but authors must correct:
- In Figure 8d, it is mentioned "Mg-sheet subsurfaces". I believe that the "subsurfaces" must be changed with "surfaces".
- In Figures 10 and 11, modify "2-time" with "2-times" and "3-time" with "3-times".
- line 247, modify "1 st" with "1st".
- line 527, modify "6.1 wt%" with "6.1 wt.%".
- line 541, modify "Figure 14" with "Fig. 14".
- line 547, modify "5.5 wt% Ni" with "5.5 wt.% Ni".
- line 555, modify "5 wt%" with "5 wt.%".
- lines 612 and 624, modify "MgH2" with "MgH2".
- line 651, modify "6.1 wt%" with "6.1 wt.%".
Author Response
- We would like to thank you very much for the critical reading of the manuscript and the important comments and kind suggestions.
- We are pleased to bring to your kind attention that all of these comments and their corresponding changes were seriously considered and achieved upon revising the manuscript.
- Besides, the “Mg-Substrates” was changed into “Mg-Surfaces”, as shown in Fig. 8a.
- Moreover, the singular form of word “time” was changed into the plural for in Figs. 10 and 11.
Again, thank you very much.
Round 2
Reviewer 2 Report
Accept.